

# A semi-automatic three-dimensional technique using a regionalized facial template enables facial growth assessment in healthy children from 1.5 to 5.0 years of age

Robin Bruggink[1,2], Frank Baan[1,2], Sander Brons[3], Tom G.J. Loonen[2], Anne Marie Kuijpers-Jagtman[4,5,6], Thomas J.J. Maal[2,7] and Edwin M. Ongkosuwito[1,8]

[1] Department of Dentistry, Radboud University Medical Center, Nijmegen, The Netherlands
[2] Radboudumc 3D Lab, Radboud University Medical Center, Nijmegen, The Netherlands
[3] Orthodontie Merwestein, Nieuwegein, The Netherlands
[4] Department of Orthodontics, University Medical Center Groningen, Groningen, The Netherlands
[5] Faculty of Dentistry, Universitas Indonesia, Jakarta, Indonesia
[6] Department of Orthodontics and Dentofacial Orthopedics, University of Bern, Bern, Switzerland
[7] Department of Oral and Maxillofacial Surgery, Radboud University Medical Center, Nijmegen, The Netherlands
[8] Amalia Cleft and Craniofacial Centre, Radboud University Medical Center, Nijmegen, The Netherlands

Corresponding author
Robin Bruggink,
orthodontie@radboudumc.nl

## ABSTRACT

**Objectives**. To develop a semi-automatic technique to evaluate normative facial growth in healthy children between the age of 1.5 and 5.0 years using three-dimensional stereophotogrammetric images.

**Materials and Methods**. Three-dimensional facial images of healthy children at 1.5, 2.0, 2.5, 3.0, 4.0 and 5.0 years of age were collected and positioned based on a reference frame. A general face template was used to extract the face and its separate regions from the full stereophotogrammetric image. Furthermore, this template was used to create a uniform distributed mesh, which could be directly compared to other meshes. Average faces were created for each age group and mean growth was determined between consecutive groups for the full face and its separate regions. Finally, the results were tested for intra- and inter-operator performance.

**Results**. The highest growth velocity was present in the first period between 1.5 and 2.0 years of age with an average of 1.50 mm (±0.54 mm) per six months. After 2.0 years, facial growth velocity declined to only a third at the age of 5.0 years. Intra- and inter-operator variability was small and not significant.

**Conclusions**. The results show that this technique can be used for objective clinical evaluation of facial growth. Example normative facial averages and the corresponding facial growth between the age 1.5 and 5.0 years are shown.

**Clinical Relevance**. This technique can be used to collect and process facial data for objective clinical evaluation of facial growth in the individual patient. Furthermore, these data can be used as normative data in future comparative studies.

## INTRODUCTION

Longitudinal normative craniofacial growth data of a large healthy population support healthcare professionals monitoring growth of the individual patient. (*Brons et al., 2012*; *Brons et al., 2019b*; *De Onis, Wijnhoven & Onyango, 2004*; *De Onis & Blössner, 1997*; *Panpanich & Garner, 1999*). Effective monitoring could indicate growth problems and underlying illnesses. Normative data can provide more insight in the diagnosis, severity, and progress of patients with known craniofacial disorders like hemifacial microsomia or cleft lip and/or palate or detect unknown disorders.

In the past, several techniques to quantitatively determine craniofacial dimensions were used. The standard technique is to use direct anthropometric measurements, which is considered the gold standard and was extensively used in the past century. Although, these measurements are inexpensive, reliable and do not need ionizing radiation or expensive equipment, they have some major drawbacks. They are very time consuming, need patient compliance and do not describe the three-dimensional (3D) superficial area (*Aung, Ngim & Lee, 1995*; *Farkas & Deutsch, 1996*). Additionally, it is not possible to remeasure the parameters in a later stadium without the use of a replica, *e.g.*, a gypsum cast of the face. Digital anthropometry using direct 3D computerized digitizers, still requires patient cooperation but takes less time and enables a basic 3D facial profile of the patient. Using a stylus, 3D coordinates of facial landmarks are collected which subsequently can be used to perform linear, angular and rough areal measurements (*De Menezes et al., 2009*; *Ferrarioet al., 2003*). In the 1930s, the cephalometric analysis was introduced using lateral skull radiographs on which linear and angular measurements can be made of facial surface landmarks, bony and dental structures (*Hofrath, 1931*; *Pittayapat et al., 2014*). However, this requires use of ionizing radiation, which makes it questionable for use in infant growth studies. Another drawback is that cephalograms represent a two-dimensional (2D) image of a 3D structure which results in loss of information. Furthermore, when the orientation of the head is not correct during image acquisition, the visualization of the structures could be skewed due to oblique projections or magnifications (*Major et al., 1996*; *Ongkosuwito et al., 2009*).

In recent years, advancements in non-contact 3D imaging techniques have made digital 3D stereophotogrammetry the preferred method to acquire quantitative information of the facial surface (*Heike et al., 2010*; *Kuijpers et al., 2014*). Using a multi-camera setup, several photos are simultaneously acquired which are then digitally stitched into a 3D surface. As the technique relies on normal photographic cameras, acquisition time is almost instant and no ionizing radiation is needed (*Heike et al., 2010*; *Thierens et al., 2018*). As these images are digital, they can easily be stored, processed and remeasured. Due to these non-invasive and near-instant characteristics, these systems are very useful when working with young children (*Brons et al., 2012*; *Heike et al., 2010*).

Unfortunately, comparison between two different 3D surfaces is not straight forward for several reasons. First, the position of the patient in two different images is mostly not identical, so they have to be aligned in a common axis; *e.g.*, using a reference frame based on the natural head position (*Brons et al., 2013*). Secondly, the surfaces do not contain the same amount of datapoints (vertices) and they are not assigned to the same features of the face. To cope with the latter, standardization is needed. The most easy and simple technique is to indicate facial landmarks, which can be compared (*Krimmel et al., 2015*; *Ogodescu et al., 2021*). However, this technique causes loss of information as it ignores most of the 3D surface area. Newer techniques often use a 3D surface template with predefined vertices for standardization. This template is fitted on the original 3D surface using algorithms like Coherent Point Drift (CPD) (*Koudelová et al., 2019*) or other non-rigid registration techniques (*White et al., 2019*). With the use of the template, growth can easily be monitored as each vertex corresponds to the same position on the face. The addition of predefined regions in the template enables quantifying growth in specific parts of the face, which is especially interesting for indicating asymmetries and abnormalities.

To date, there are little normative data collected regarding 3D facial growth in young children, especially before the age of 3 years (*Brons et al., 2012*; *Kočandrlová et al., 2021*; *Sforza et al., 2018*). Normative data are essential for comparative studies to indicate altered facial development or treatment outcomes in children with craniofacial malformations. The main goal of this study was to develop a semi-automatic technique to evaluate growth of a healthy population ranging between 1.5–5.0 years of age, using a regionalized facial template. Furthermore, the normative growth data for this population was calculated.

## MATERIALS & METHODS

### Subjects

The subjects included in this prospective longitudinal cohort study were derived from an earlier study of *Brons et al. (2019b)*. Healthy infants were included between April 2007 and September 2010 at the Maternity Clinic at the Radboudumc, Nijmegen, The Netherlands and the Regional Health Services (GGD Gelderland-Zuid, Nijmegen, The Netherlands). All infants were from Caucasian birth, born full term (38+ weeks) and had no first- to third degree relatives with craniofacial disorders.

The study protocol was approved by the medical ethical commission of the Radboud University Medical Centre (CMO 2007/163). All subjects' parents provided written informed consent prior to inclusion in the study.

### Data acquisition

Acquisition of the 3D facial images was performed with the 3dMDCranial system (3dMD Ltd., Atlanta, GA, USA) with a two-pod configuration. This system was located in a room without windows and had consistent ambient light. Shutter time was 1.5 ms, illumination was achieved using multiple xenon flash units and system calibration was performed every day. When possible, hair nets or ties were used to minimize the interference with hair. If the child was not able to sit still on its own, the parents were asked to hold the child with extended arms. Images were made at 1.5, 2.0, 2.5, 3.0, 4.0 and 5.0 years of age and were

a)                                        b)

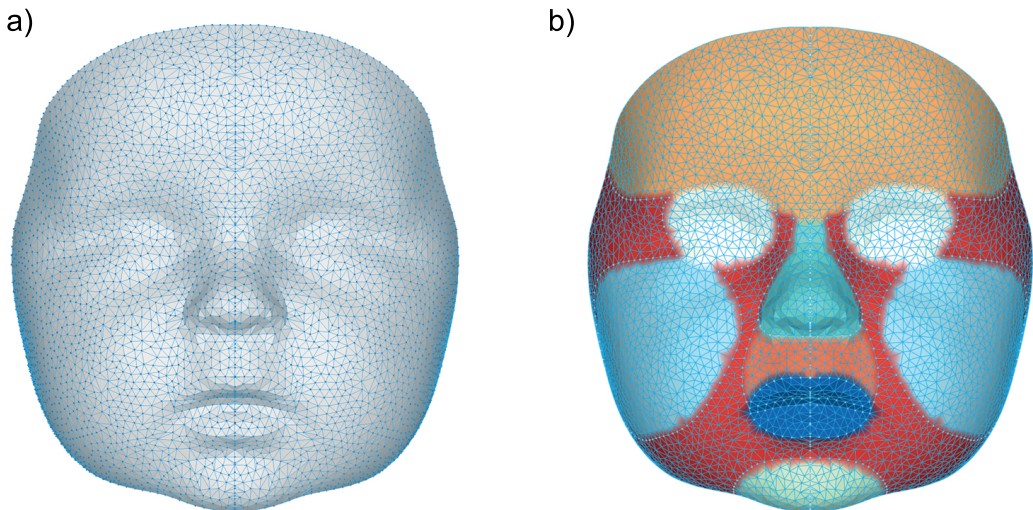

Figure 1 **Visualization of the general face template.** (A) The data points representing each vertex within the general face template, (B) the separate regions on the general face template represented by the following colors: orange (forehead), white (eyes), cyan (nose), dark orange (subnasal), dark blue (mouth), light blue (cheeks), light green (chin) and red (other).

reconstructed to 3D facial images using 3dMDPatient 4.0 software (3dMD Ltd., Atlanta, GA, USA). The resulting meshes were exported to a Wavefront OBJ file.

Before study inclusion, the images were visually checked. Images were included if all the following criteria were met: (1) neutral facial expression, (2) the presence of minimal one of the two tragi, (3) acceptable image quality (no stitching errors, smooth surface *etc.*), and (4) no large holes or gaps within the facial region.

## General face template

A general face template based on five randomly chosen faces, was used to process the 3D images. The vertices in this template were remeshed and reduced to a uniform spread mesh of 3,602 datapoints (Fig. 1). Several facial regions were manually annotated on this template to enable distinguishing growth on a regional scale (*Maal et al., 2011*). These regions include; the cheeks, forehead, nose, subnasal area and the chin. Although the eye and mouth area were also selected, these were not included in any of the analyses, because the 3D camera cannot accurately determine reflective or moist surfaces like the eye white and saliva.

## 3D landmarking

The 3D images were imported into 3DMedX® (v1.3.0.0, 3D Lab Radboudumc, Nijmegen, The Netherlands). Several facial landmarks (*Urbanová, 2016*) needed for alignment were manually placed and exported to a comma separated values (CSV) file using the batch analysis plugin of 3DMedX®. The description of the landmarks is given in Table 1 and visualized in Fig. 2.

**Table 1 Definition of the landmarks used to orient and prepare the 3D images before analysis (*Urbanová, 2016*).**

| Landmark | Bilateral | Determination | Description |
|---|---|---|---|
| Tragus (tr) | Yes | Manual | The most lateral point on the tragus |
| Exocanthion (ex) | Yes | Manual | Lateral canthus of the eye |
| Endocanthion (en) | Yes | Manual | Medial canthus of the eye |
| Pupil (p) | Yes | Automatic | The center of the eye, calculated as the midpoint between the endo- and exocanthion |
| Pronasale (prn) | No | Manual | Most anterior point on the midline of the nose |
| Cheilion (ch) | Yes | Manual | Lateral point on the labial commissure |
| Pupil reconstructed point (prp) | No | Automatic | Landmark that is automatically determined by averaging both end- and exocanthions |

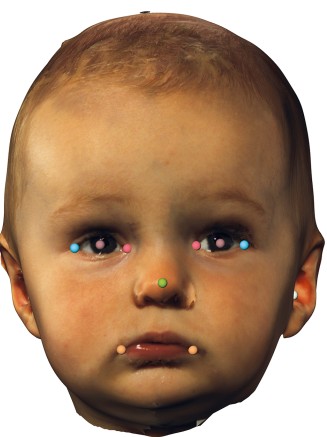 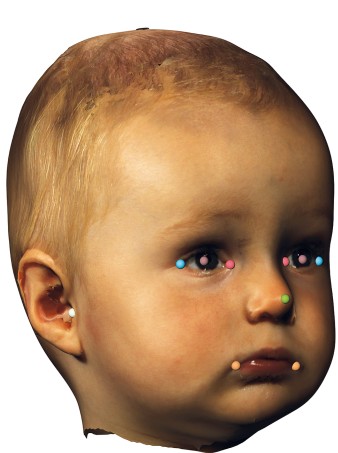

○ Exocanthion (ex)
● Pupil (p)
● Endocanthion (en)
● Pronasale (prn)
○ Cheilion (ch)
○ Tragus (tr)

**Figure 2 Visualization of the used landmarks to orient and prepare the data before analysis.** The definitions are given in Table 1.

## 3D face alignment

To create an average face, the 3D images must have the same orientation and position in space. For this, an adapted version of the reference frame of *Brons et al. (2013)* implemented in Matlab (MATLAB 2020b The MathWorks, Inc., Natick, Massachusetts, USA) was used. Alignment was performed in four steps; (1) The image was centered on the pupil reconstructed point (PRP), which was calculated by averaging both endo- and exocanthia landmarks. (2) The sagittal plane was placed perpendicular to the canthal line determined by the vector between both exocanthia. (3) The horizontal plane was placed through the pupil reconstructed point perpendicular to the line defined from tragus to exocanthion and the sagittal plane. When both tragi were present, the average of both was taken. The

**Table 2 The number of included 3D images and their characteristics.** To maximize the number of included images, only the forehead region was excluded when hair was visible at that location.

| Age Group | 1.5y | 2.0y | 2.5y | 3.0y | 4.0y | 5.0y | Total |
|---|---|---|---|---|---|---|---|
| Selected images | 59 | 50 | 39 | 31 | 32 | 21 | 232 |
| Excluded meshes | 0 | 0 | 0 | 5 | 0 | 0 | 5 |
| Excluded foreheads | 7 | 5 | 7 | 4 | 4 | 2 | 29 |
| Total meshes (including forehead) | 59 (52) | 50 (45) | 39 (32) | 26 (22) | 32 (28) | 21 (19) | 227 (198) |
| Target age in months | 18.0 | 24.0 | 30.0 | 36.0 | 48.0 | 60.0 | – |
| Mean age ± SD included meshes in months | 18.0 ± 1.0 | 24.4 ± 0.5 | 30.4 ± 1.1 | 36.6 ± 0.5 | 48.5 ± 2.3 | 60.7 ± 2.0 | – |

coronal plane was placed though the tragus and perpendicular to the other two planes. A rigid iterative closed point algorithm between the face and a template was used to finetune the pitch of the face.

## Mesh processing

Data was prepared for analysis by manually correcting small defects and holes in the mesh. After this, the 3D images were remeshed using Meshlab (version 2020.v7) (*Cignoni et al., 2008*) to obtain a uniform spaced mesh with a target edge length of 1.5 mm.

To assess the facial dimensions, a semi-automatic analysis was created using Matlab and parts of the MeshMonk library (*Matthews et al., 2021*; *White et al., 2019*). This analysis consists out of multiple steps, as shown in Fig. 3: (A) The general face template was roughly aligned on each image using the Procrustes algorithm with both endocanthia, cheilions and, the pronasale landmarks as reference. Afterwards, a rigid iterative closest point registration was used to further align the template. (B) To minimize the artefacts and to clean up the 3D surface, datapoints outside the template region were removed. (C) Using a non-rigid registration technique described by White and Matthews et al. (*Matthews et al., 2021*; *White et al., 2019*), the registered template was morphed towards the contours of the facial mesh. (D) Finally, a ray-tracing algorithm was used to project each of the vertex's locations of the template on the facial mesh. The result is a uniform indexed mesh whose vertices can directly be compared with the vertices in other prepared meshes. When hair was present in the resulting mesh, the forehead region was removed in further analyses. With all resulting meshes, average faces were constructed for each age group.

Inter-surface distances were calculated between each average face of the consecutive age groups, indicating growth between the age groups. The inter-surface distance was measured by calculating the Euclidean distance between each corresponding vertex of the first and the second model. The inter-surface distances were sorted into the spatial regions of interest present in the general template to compare regional changes during the child's development. All results were converted to millimeter growth per six months to enable comparison.

## Statistical analysis

The growth of the face and regions are presented as the mean, standard deviation of the mean, and the percentiles p5-p95 of the inter-surface distances between each consecutive
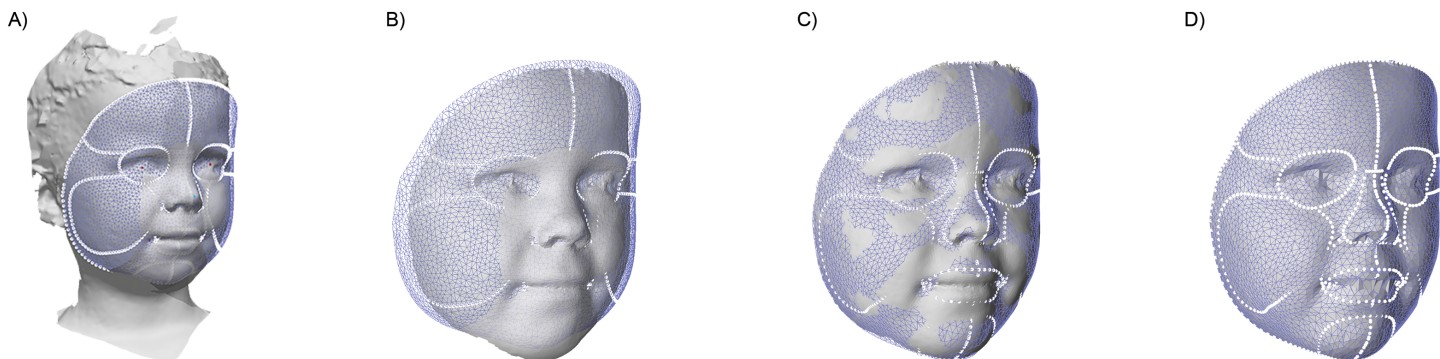

**Figure 3** **The four steps of the image preparation.** (A) Using the manual clicked landmarks, a general face template is roughly aligned on the 3D image. (B) Excess data is removed by using the registered template, retaining only the face. (C) A non-rigid registration algorithm is used to morph the registered template on the face. (D) The template is projected back on the original image.

age group. Visual assessment of the facial growth was performed by using color-coded distance maps, which help to indicate local changes.

The reliability of the analysis was tested by repeating the mesh preparation and analysis procedure for 20 randomly selected 3D images. This was repeated after a minimum of two weeks by the same observer to determine the intra-operator variability. A second observer repeated this as well to assess the inter-operator variability. Systematic differences were calculated using the paired sample $t$-test and the Dahlberg coefficient was used to assess the variance.

All statistical analyses were performed using SPSS 26 (IBM Corp. Released 2018. IBM SPSS Statistics for Windows, Version 26.0 Armonk NY, IBM Corp), the significance level was set at $p < 0.05$.

# RESULTS

A total of 75 healthy children were included in this study, 36 males (48%) and 39 females (52%). A total of 232 3D images were obtained and grouped at 1.5, 2.0, 2.5, 3.0, 4.0 and 5.0 years after birth. After the analysis, 5 images were excluded due to topological mesh errors. This was caused as the facial image did not completely cover the template, and therefore the template was warped around the edge of the image. Furthermore, in 29 processed meshes the forehead was excluded as hair was present within the facial area. Finally, a total of 227 meshes were analyzed in this study (Table 2).

## Reliability of the method

Reliability tests showed no significant differences within and between observers ($p = 0.50$ and 0.44, respectively). Within observers, a mean difference of 0.17 mm ($\pm 0.71$ mm) was found with a confidence interval between $-0.68$ and 0.50 mm. Between observers the mean difference was $-0.08$ mm ($\pm 0.63$ mm) with a confidence interval between $-0.37$ and 0.22 mm (Table 3).

**Table 3** The intra- and inter-operator differences between the calculated distance maps.

| Parameter | Mean difference (mm) | Std (mm) | 95% CI (mm) | $p$ | Dahlberg coefficient |
|---|---|---|---|---|---|
| Intra-operator | 0.17 | 0.71 | [−0.16 to 0.50] | 0.29 | 0.50 |
| Inter-operator | −0.08 | 0.63 | [−0.37 to 0.22] | 0.59 | 0.44 |

**Notes.**
Std, Standard Deviation; 95% CI, 95% Confidence Interval.

## Growth of the face and its regions

The facial growth between each consecutive group is described for the whole face and the individual regions in Table 4. Corresponding color-coded distance maps are shown in Fig. 4 and provide more detailed information about the location of the growth in this population.

The first period between 1.5 and 2.0 years is the period in which facial growth is present in all regions and is the largest of all time periods. The average growth of the full face is 1.50 mm and it is most prominent in the (sub)nasal and chin area.

Between 2.0 and 2.5 years, the total amount of growth of the full face decreased towards 1.13 mm. This decrease was especially present around the maxillary and mandibular regions while the largest amount of growth occurred in the regions around the nose and chin with values of 1.43 and 2.09 mm, respectively.

In the interval from 2.5 to 3.0 years, the overall facial growth stabilizes, retaining an average growth of 1.07 mm. An exception is found in the areas around the chin where the growth still declined, and the (sub)nasal area where a slightly increased velocity is seen.

In the period from 3.0 to 4.0 years, stabilization of the overall facial growth velocity continues. The largest amount of growth can be seen in the subnasal area and around the chin and lower cheeks, where some rebound of growth is present. The largest drop of growth velocity can be seen in the nasal area, where it decreased about 30%.

Finally, in the age group between 4.0 and 5.0 years the trend of decreased growth continues, halving the total amount of growth to 0.54 mm for the full face. This period is characterized with the lowest growth velocities present within the total measured time frame. Although the spread of growth seems uniform over the complete face, a hotspot can still be seen around the pronasale area (Fig. 4)

## DISCUSSION

The aim of this study was to develop a semi-automatic technique to evaluate normative facial growth and to describe and visualize the growth between the age of 1.5 and 5.0 years in healthy children using surface data acquired with a 3D stereophotogrammetry setup.

The accuracy and reproducibility of the used 3D stereophotogrammetry setup is thoroughly tested (*Boehnen & Flynn, 2005*; *Verhulst et al., 2018*). Verhulst et al. reported a reproductivity of 0.18 mm (±0.15 mm) for the setup used in our study, which can be considered clinically irrelevant. Furthermore, analyses performed on 3D images are comparable with the ones using direct anthropometry or CBCT only falling short on

**Table 4  Growth of the full face and its separate regions from 1.5 to 5.0 years of age.** Growth is displayed in mm per half year.

| Region | Group (years) | Mean growth (mm/0.5y) | std (mm) | P5–95% (mm) |
|---|---|---|---|---|
| Full Face | 1.5–2.0 | 1.50 | 0.54 | [0.76 to 2.60] |
| | 2.0–2.5 | 1.13 | 0.35 | [0.58 to 1.77] |
| | 2.5–3.0 | 1.09 | 0.23 | [0.76 to 1.51] |
| | 3.0–4.0 | 1.07 | 0.38 | [0.54 to 1.78] |
| | 4.0–5.0 | 0.54 | 0.18 | [0.29 to 0.89] |
| Nose | 1.5-2.0 | 1.85 | 0.31 | [1.39 to 2.28] |
| | 2.0–2.5 | 1.43 | 0.22 | [1.05 to 1.72] |
| | 2.5–3.0 | 1.58 | 0.18 | [1.28 to 1.82] |
| | 3.0–4.0 | 1.08 | 0.13 | [0.89 to 1.30] |
| | 4.0–5.0 | 0.95 | 0.18 | [0.62 to 1.17] |
| Forehead | 1.5-2.0 | 1.25 | 0.29 | [0.76 to 1.72] |
| | 2.0–2.5 | 1.01 | 0.21 | [0.60 to 1.33] |
| | 2.5–3.0 | 0.97 | 0.15 | [0.77 to 1.27] |
| | 3.0–4.0 | 0.80 | 0.21 | [0.47 to 1.12] |
| | 4.0–5.0 | 0.39 | 0.08 | [0.24 to 0.51] |
| Subnasal | 1.5-2.0 | 2.06 | 0.20 | [1.68 to 2.36] |
| | 2.0–2.5 | 1.26 | 0.20 | [0.93 to 1.54] |
| | 2.5–3.0 | 1.41 | 0.13 | [1.24 to 1.66] |
| | 3.0–4.0 | 1.28 | 0.15 | [1.06 to 1.56] |
| | 4.0–5.0 | 0.71 | 0.18 | [0.47 to 1.04] |
| Chin | 1.5-2.0 | 2.88 | 0.18 | [2.62 to 3.13] |
| | 2.0–2.5 | 2.09 | 0.21 | [1.72 to 2.37] |
| | 2.5–3.0 | 1.28 | 0.24 | [0.84 to 1.60] |
| | 3.0–4.0 | 1.81 | 0.12 | [1.63 to 2.03] |
| | 4.0–5.0 | 0.84 | 0.07 | [0.71 to 0.92] |
| Cheeks | 1.5-2.0 | 1.31 | 0.30 | [0.80 to 1.81] |
| | 2.0–2.5 | 1.07 | 0.22 | [0.66 to 1.41] |
| | 2.5–3.0 | 1.06 | 0.15 | [0.77 to 1.32] |
| | 3.0–4.0 | 1.06 | 0.25 | [0.69 to 1.50] |
| | 4.0–5.0 | 0.56 | 0.08 | [0.42 to 0.69] |

Notes.
  std, Standard Deviation; P5–95%, 5–95% percentile.

some bony landmarks as they are not clearly visible or palpable, *e.g.*, the gonion landmark. (*Aynechi et al., 2011*; *Dindaroğlu et al., 2016*; *Flügge et al., 2013*; *Lübbers et al., 2010*; *Plooij et al., 2009*). The reliability analyses used in this study showed a high agreement between and within observers as only small, non-significant, differences were found. These differences are most likely due to the manual landmarking as the rest of the analysis is automated. When considering the reference frame, the identification of the tragus landmark has the most influence on the outcome as it is used as the center of antero-posterior growth. Meaning that an error in this antero-posterior position of this landmark will result in a direct offset for the complete face. Automatic landmarking could solve this problem, but

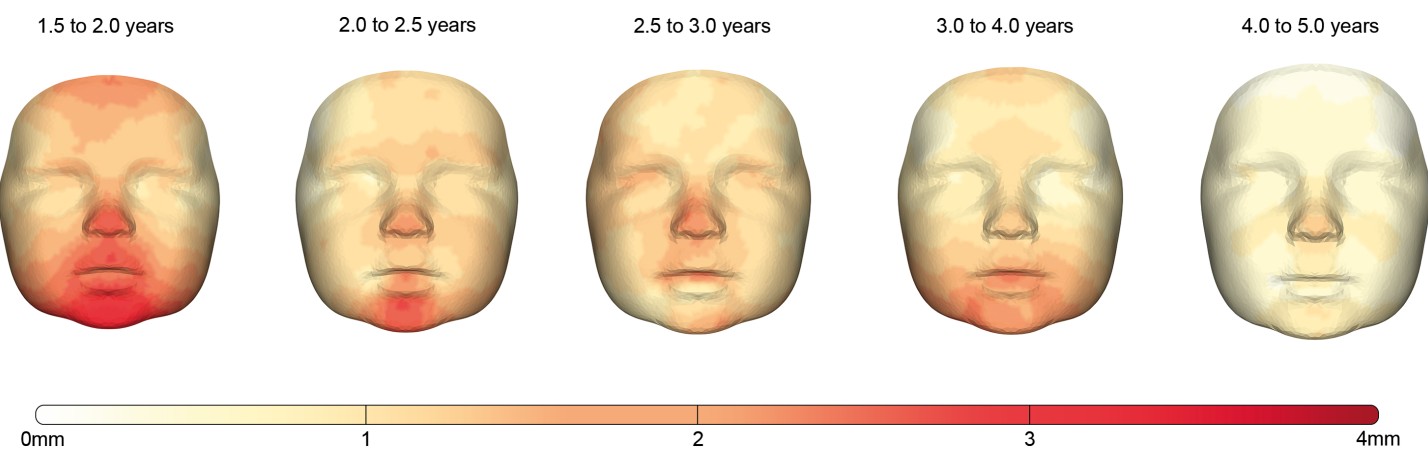

**Figure 4  The growth between each age group visualized by color-coded distance maps.** The intensity of the red color determines the degree of growth indicated by the color scale. As only growth was present, the negative part of the scale was removed for clarity.

it would be difficult to determine this tragus landmark due to its location at the edge of the image. The two-pod configuration is not able to acquire many reference points at the edges, resulting in a less accurate and low-density mesh. However, it would be feasible to automatic determine the initial alignment of the template, as the rigid registration only requires a rough estimation. After the registration process, the endo- and exocanthia landmarks could be extracted from the template which can then be used to apply the reference frame. Techniques like curvature analysis (*Creusot et al., 2013*) and artificial intelligence (*Zhang et al., 2010*) could help with this.

A general face template enables direct cross-sectional and longitudinal comparison between different 3D images as each datapoint corresponds to the same location on all of the faces, *e.g.*, a datapoint on the tip of the nose from an image corresponds with a datapoint on the tip of the nose in another image (*Brons et al., 2019b*). With this knowledge, the facial regions could be transferred from the template to the 3D image, which can quantify local changes. The general face template was created using an average face which was based only on a few selected images. This was done to preserve the original facial contours and features which are important for accurate registration.

Although the amount of facial 3D images in research databases is rising, there is still only little reference information on 3D facial growth in young children (*Kuijpers-Jagtman, 2012*; *Weinberg et al., 2016*). Therefore, no reliable sample size calculations could be made in advance for the number of required subjects needed for creating average faces. Some studies investigating facial growth in children with comparable templating techniques were found. A study of *Kočandrlová et al. (2021)* used a total of 42 healthy preschool children to determine the facial dimensions annually between 3 and 6 years. *Koudelová et al. (2019)* investigated longitudinal growth in older children and included around 45 healthy subjects per group in the age range of 7 to 17 years.

In this study we have adapted the reference frame used in the previous studies which used the angle between the PRP and tragus to determine the pitch of the face (*Brons et*

*al., 2019a*; *Brons et al., 2019b*). Using the latter method resulted in a large variation of the facial orientations. As the reported inter-operator variability was low, other influences could be the culprit. One possibility is that the tragal height between different subjects is not comparable, *i.e.*, there is a large vertical variation of the tragus on the facial surface. Although there are many anthropometrical studies describing the external ear, these are often limited to the dimensions rather than the location on the face (*Kalcioglu et al., 2003*; *Muteweye & Muguti, 2015*). Another possibility is that the angle between the tragus and PRP changes during development as the growth velocity is not synchronic around the two landmarks. Therefore, the use of a general face template to determine the pitch of each face to guarantee a representable and reproducible orientation was chosen. However, this method can partially average out the actual pitch of the individual patient's face.

The clinical results indicate that the largest amount of facial growth is present during the first period ranging from 1.5 to 2.0 years after birth. In general, the growth declines afterwards until it is at a third of its initial velocity at 5.0 years. Although each individual region expresses different growth velocities, it generally follows the decrease present in the full face. Visual assessment of the distance maps shows particular more growth at the tip of the nose in comparison with the nasal region. This indicates that the overall shape of the nose is changing the most during early development, where the caudal part of the nose is getting more prominent per ratio. During the period between 1.5 and 2.5 years, development is more dominant at the caudal part of the chin. This is most likely the effect of the elongation of the face and the development of the jaws during this timeframe. This is supported by the research of *Liu, Behrents & Buschang (2010)* who found that most of the mandibular growth and especially the mandibular length, took place during the first 2 years of life. Furthermore, child's cooperation may have influenced the location of the chin, *e.g.*, not fully closing the mouth positions the chin caudally.

In the earlier study of *Brons et al. (2019b)* normative facial growth within the first year after birth was assessed. When comparing the growth between 6–12 months in their study with the growth from 18 to 24 months in the present study, a large decrease in growth velocity can be seen. The growth velocity for the full face decreased from 5.10 to 1.50 mm, remaining only a third of its original value. This decline is seen in the other regions as well; the growth velocity of the nose decreased from 5.90 to 1.85 mm, the chin from 6.40 to 2.88 mm, the forehead from 6.10 to 1.25 mm and finally the cheeks from 3.70 to 1.31 mm. The largest decrease of growth velocity is seen in the forehead area. This can be explained by the high postnatal growth rate of the neurocranium due to sutural growth (*Frassanito et al., 2019*; *Krimmel et al., 2015*). Most of the neurocranial growth occurs in the first year after birth, which is estimated from 25% of its adult size at birth to 65% after 12 months (*Frassanito et al., 2019*; *Kamdar, Gomez & Ascherman, 2009*). After the first year of life the sagittal cranial sutures start to close, and therefore the expansion of the neurocranium is limited (*Krimmel et al., 2015*).

Overall, the total decline of growth velocity in this study is comparable with the general growth standards published by the World Health Organization, where the largest growth velocities can be seen in the first 12 months after birth and strongly decline afterwards (*World Health Organization & Nutrition for Health and Development, 2009*).

This is especially true for the head circumference, which is best comparable with the facial growth described in the present study as they both describe the outward growth of the head/face.

With the current number of included patients, the average facial growth in children was described. However, due to a large variation in growth between children, it is recommended to continue longitudinal cohort studies of healthy children of a variety of backgrounds and ethnicities to strengthen the results of future growth models. Moreover, this is needed as the data in this study were derived from a single uniform research population, which makes the results only applicable for this Caucasian population. Monitoring individual growth to create prediction models (*Bruggink et al., 2019*) was not possible due to missing longitudinal data because of missed appointments and loss-to-follow-up which decreased the sample size in the older age groups. This is a common problem in any longitudinal study. Furthermore, in the past it was thought that sexual dimorphism primary arises around puberty. But recent studies suggest that this could be present earlier in life (*Kočandrlová et al., 2021*). The research of *Matthews et al. (2016)* shows that some degree of sexual dimorphism is already present at the age of 1 year. For all reasons mentioned longitudinal cohort studies are required with a large sample size, which are expensive and take a long timeframe before any results become available. This seems only possible in a concerted action by collaborations in international research consortia.

Another limitation is poor patient cooperation of young children. Non-neutral and involuntary facial expressions can have a major influence on the assessments (*Brons et al., 2019a*), and therefore various 3D images could not be included in this study. Acquiring suitable 3D images in young children with the current 'one picture at a time' setup is challenging and requires experienced photographers (*Brons et al., 2019a*; *Kočandrlová et al., 2021*). Currently, imaging setups are available that include the temporal dimensions when acquiring surface information. These 4D systems make it possible to record the subject during a short time frame and create 3D images based on the most suitable frames afterwards. This prevents repeating the acquisition many times to acquire a suitable 3D image, making it faster and more feasible to use 3D imaging in young children.

## CONCLUSIONS

This study has presented a semi-automatic technique to indicate facial growth. With this technique the average faces and facial growth of children aged between 1.5 and 5.0 years could be investigated. When the sample size increases, the resulting normative data can be used to evaluate facial growth and the effect of treatment in individual patients with orofacial disorders. In future research, separate average faces for boys and girls should be implemented for increased accuracy.

## ACKNOWLEDGEMENTS

The authors are grateful to all the parents and their children participating in this study. Special hanks go out to Jene Meulstee for his preparatory work used in this study.

### Funding

The authors received no funding for this work.

### Competing Interests

Robin Bruggink is a developer for 3DMedX® which is a commercial research package used in this study. Anne Marie Kuijpers-Jagtman is an Academic Editor for PeerJ. Sander Brons is an orthodontist at Orthodontie Merwestein. The other authors declare that they have no competing interest.

### Author Contributions

- Robin Bruggink conceived and designed the experiments, performed the experiments, analyzed the data, prepared figures and/or tables, authored or reviewed drafts of the article, and approved the final draft.
- Frank Baan performed the experiments, analyzed the data, authored or reviewed drafts of the article, and approved the final draft.
- Sander Brons conceived and designed the experiments, authored or reviewed drafts of the article, and approved the final draft.
- Tom G.J. Loonen conceived and designed the experiments, analyzed the data, authored or reviewed drafts of the article, and approved the final draft.
- Anne Marie Kuijpers-Jagtman conceived and designed the experiments, authored or reviewed drafts of the article, and approved the final draft.
- Thomas J.J. Maal conceived and designed the experiments, authored or reviewed drafts of the article, and approved the final draft.
- Edwin M. Ongkosuwito conceived and designed the experiments, authored or reviewed drafts of the article, and approved the final draft.

### Human Ethics

The following information was supplied relating to ethical approvals (i.e., approving body and any reference numbers):

Medical ethical commission of the Radboud University Medical Centre

### Data Availability

The MATLAB code, and results, manual clicked landmarks and average faces are available in the Supplementary File.

The Meshmonk Library is available at GitHub: https://github.com/TheWebMonks/meshmonk.

The data that support the findings of this study are available on request to the Privacy Department of Radboud: privacy@radboudumc.nl. The data are not publicly available due to the identifiable nature of 3D images which could compromise the privacy of vulnerable research participants.

## Supplemental Information

Supplemental information for this article can be found online at http://dx.doi.org/10.7717/peerj.13281#supplemental-information.

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
