# Peer review of "A semi-automatic three-dimensional technique using a regionalized facial template enables facial growth assessment in healthy children from 1.5 to 5.0 years of age"

_PeerJ, doi:10.7717/peerj.13281_

## Round 0.1 · original submission · Minor Revisions

Thank you for the submission. Please find the comments below from the reviewers and do the needful changes. Please submit a detailed response to the reviewer's comments. I look forward to your revision.

Reviewer 1 ·

Basic reporting

The current work deals with a topic of interest for all clinicians working with aging young subjects. The manuscript is well written and provides original data concerning the growth of very young children which are not so studied due to many difficulties of the methodology used.

Experimental design

The methodology used is accurate, updated and reliable.
I would only suggest some changes to improve the paper as follows:
Introduction
• Please complete the panoramic of the methodology available for anthropometric measurements of the face by citing direct 3D computerized digitizers.
• At line 105 add a reference about the normative data already present in the literature.
Materials and methods
• In data acquisition some additional info concerning the preparation of the child such as positioning, use of a hair tie, explanations, etc.. could be useful for eventual reproduction of the recording and/or for discussion of the limitation and further comparisons.
Results
• Please specify which topological mesh errors were found, causing some images’ exclusion.
Discussion:
• Please add a reference when discussing in landmark location and artifacts, see line 250. topological mesh errors were found, causing some images’ exclusion.

Validity of the findings

The system and the analyses here seem to be reliable and generate sound results that need further confirmation by means of future investigations in larger samples. Therefore, I think that the current manuscript could be considered for publication.

·

Basic reporting

very interesting article, well written and excellently planned.
I suggest the following points:
1) better specify the definition of the analysis areas defined by the template and why defined in this way
2) propose reasons for the curious result of lower variability in the inter-operator repeatability test compared to intra-operator
3) propose solutions to automate even the first manual alignment. In previous studies, landmark alignment was used or less average distance of selected areas. ears?
J Prosthodont. 2014 Jul; 23 (5): 347-52.
Int Int J Oral Maxillofac Surg. 2012 Nov; 41 (11): 1344-9
4) correct citations in the text that are not always correctly formatted
5) I suggest the addition and discussion of the following recent publication, with a different purpose but with an overlapping methodology Children 2022, 9 (2), 187

Experimental design

no structural suggestions

Validity of the findings

ok

Additional comments

well done

---

## Round 0.2 · accepted · Accept

Thank you for the revision. I am satisfied with the manuscript now and am happy to accept the paper.